# Photosynthetic Physiological Basis of No Tillage with Wheat Straw Returning to Improve Maize Yield with Plastic Film Mulching in Arid Irrigated Areas

**DOI:** 10.3390/plants12061358

**Published:** 2023-03-17

**Authors:** Yao Guo, Hong Fan, Pan Li, Jingui Wei, Hailong Qiu

**Affiliations:** 1College of Life Sciences, Northwest Normal University, Lanzhou 730070, China; 2State Key Laboratory of Aridland Crop Science, Gansu Agricultural University, Lanzhou 730070, China

**Keywords:** maize, straw management, grain yield, photosynthetic physiological traits, arid regions

## Abstract

Surface mulch is an efficient plant production technique widely used in arid and water-scarce areas. In this study, a field experiment was conducted to determine whether plastic film combined with wheat straw returning could boost grain yield of maize via optimizing photosynthetic physiological characteristics and coordinating yield components. The results showed that no tillage with wheat straw mulching and straw standing treatments had better regulation on photosynthetic physiological characteristics and had a greater impact on the increase in grain yield than conventional tillage with wheat straw incorporation and without wheat straw returning (the control treatment) in plastic film-mulched maize. Meanwhile, no tillage with wheat straw mulching had a relatively higher yield than no tillage with wheat straw standing through better regulation of photosynthetic physiological characteristics. No tillage with wheat straw mulching decreased the leaf area index (LAI) and leaf area duration (LAD) of maize before the VT stage and maintained higher LAI and LAD after the VT stage, which effectively regulated the growth and development of maize at early and late stages of development. From VT to R4 stage of maize, no tillage with wheat straw mulching had greater chlorophyll relative content, net photosynthetic rate, and transpiration rate by 7.9–17.5%, 7.7–19.2%, and 5.5–12.1% than the control, respectively. In addition, leaf water use efficiency was increased by 6.2–6.7% from the R2 to R4 stage of no tillage with wheat straw mulching in comparison to the control treatment. Thus, no tillage with wheat straw mulching had a greater grain yield of maize by 15.6% than the control, and the high yield was attributed to the synchronous increase and cooperative development of ear number, grain number per ear, and 100-grain weight. Collectively, no tillage with wheat straw mulching had a positive effect on regulating the photosynthetic physiological traits and can be recommended to enhance the grain yield of maize in arid conditions.

## 1. Introduction

Water resource shortage has become the core problem affecting global agricultural production [1,2]. Globally, arid and semi-arid areas with water shortages account for more than 40% of the total arable area and feed more than 30% of populations around the world [3,4]. Agricultural production in arid areas where water resources are scarce plays an important role in the global food supply. The oasis-irrigated regions in northwest China are the main areas of grain production, and extreme climate and water shortage are the main limiting factors to agricultural production in this region [5,6].

Photosynthesis in plants is a very complicated process, involving biochemical, electrochemical, and physicochemical reactions [7,8]. Soil hydrothermal characteristics are the key ecological factors that regulate plant photosynthetic performance and affect its production [9,10]. Many publications have demonstrated that water scarcity and high soil temperature decreased leaf area index (LAI), leaf area duration (LAD), and chlorophyll relative content for green leaves by inhibiting root action of crops, thus reducing net photosynthetic rate (Pn) and transpiration rate (Tr) and affecting the grain filling [11]. Soil drought inhibited the photochemical activity of PSII by reducing leaf water content and LAI for plants, which resulted in the decrease in Pn and Tr, and affected the yield formation [11,12]. Previous studies have shown that photosynthesis of plants from flowering to maturing stage accumulates higher than 65% of the carbohydrate was used for grain formation [8,13], it is very important to keep relatively high photosynthetic capacity in the late growing period for yield increase and stability. So, it is of great significance to improve grain yield by optimizing cultivation measures to create suitable soil ecological environment and maintain higher photosynthetic capacity of plants in late growth period.

The photosynthetic capacity of plants is determined by their own genetic characteristics and surrounding environmental conditions [7,14]. The influence of environmental conditions on plant photosynthetic efficiency was further enhanced in arid areas with water shortages and serious desertification [5,15]. The selection of tillage practice and mulch materials is the key strategy to affect the aboveground photosynthetic performance of plants by adjusting the soil’s ecological environment [16]. Straw returning has been widely used in semi-arid and arid areas where there is a scarcity of water resources due to its advantages of harvesting and maintaining soil moisture [17,18]. Straw returning can achieve high and stable yield and efficient water utilization by reducing ineffective evaporation of soil water and enhancing soil water retention and reducing soil heat loss to air [19,20,21]. Straw returning was beneficial to promoting cooperative growth of above- and below-ground crops and enhancing crop productivity by improving the hydrothermal microenvironment of topsoil [22,23]. However, straw returning combined no tillage sometimes affects normal growth and development by reducing photosynthetic performance, which is because the lower soil temperature inhibits the root activity of plants [24,25]. Therefore, it is very urgent to study an effective straw-returning method to boost crop production by regulating the photosynthetic physiological characteristics in arid environments.

Maize (*Zea mays* L.) is one of the most important grain crops in the semi-arid region of northwest China, and the cultivation of warm-loving maize often relies on plastic film mulching due to water shortage and high evaporation [26,27]. However, plastic film mulching often causes extremely high soil temperatures in the root distribution zone at the reproductive growth period of maize plants, which can cause senescence of roots or leaves for plants and decreased yielding ability [28,29]. Considering the above issues, we developed a ‘spring wheat–maize rotation’ system, integrating the plastic film coupled with various wheat straw-returning approaches into maize production. The objectives were to determine (1) the responses of photosynthetic physiological characteristics of maize with plastic film mulching to previous wheat straw management practices; (2) the effects of previous wheat straw management practices on grain yield and yield components of maize with plastic film mulching. We hypothesized that previous wheat straw returns could improve crop yield by optimizing photosynthetic physiological characteristics and further coordinating yield composition for maize. This hypothesis was tested using a field experiment that could select a wheat straw returning way that is useful and suitable for maize production in arid, semi-arid, and other areas with similar climatic and ecological conditions.

## 2. Results

### 2.1. Dynamics of Photosynthetic Source for Maize with Plastic Film Mulching under Various Wheat Straw-Returning Approaches

#### 2.1.1. Leaf Area Index

Previous wheat straw returns had a significant effect on optimizing the dynamics of the leaf area index (LAI) of maize with plastic film mulching (Figure 1). On average, in 2010 and 2012, the LAI of three wheat straw-returning treatments was 21.9% lower than that of CT before the VT stage of maize, NTSM and NTSS significantly decreased LAI of maize by 14.1% and 18.2% compared to CTS and 24.9% and 28.4% compared to CT, respectively. However, after the VT stage, NTSM and NTSS treatments increased the LAI of maize by 22.7% and 23.9% compared to CT, NTSM had the greatest effect on the increase in LAI for maize, increased by 16.7% compared to CTS and 32.0% compared to CT. In terms of average LAI during the entire growth period of maize, the LAI of the three wheat straw-returning treatments was 13.0% higher than that of CT, the LAI of NTSM and NTSS was 11.4% and 5.5% higher than that of CTS and 19.2% and 12.9% higher than that of CT, and it was 5.6% with NTSM higher than that of NTSS.

#### 2.1.2. Leaf Area Duration

Similar to LAI, previous wheat straw-returning treatments decreased leaf area duration (LAD) by 17.1% compared to CT, and NTSM and NTSS treatments decreased LAD by 6.9% and 14.1% compared to CTS and 14.7% and 21.4% compared to CT, respectively, before the VT stage of maize (Figure 2). However, after the VT stage, previous wheat straw-returning treatments had greater LAD of maize by 16.6% than CT, and NTSM and NTSS had greater LAD of maize by 14.5% and 8.4% than CTS and 26.2% and 19.4% than CT. Over the study period, the total LAD under the three previous wheat straw-returning treatments was greater by 12.7% than that under CT, respectively. NTSM obtained the greatest total LAD and increased by 11.1% over CTS and 18.5% over CT.

### 2.2. Photosynthetic Physiological Characteristics of Various Wheat Straw-Returning Approaches to Green Leaves in Maize with Plastic Film Mulching

#### 2.2.1. Chlorophyll Relative Content

Wheat straw returning had a greatly significant influence on chlorophyll relative content (SPAD) of green leaves for maize with plastic film mulching (Figure 3). Compared to CT, NTSM and NTSS decreased SPAD for maize by 5.3% and 6.3% at the V8 stage, and NTSS decreased SPAD for maize by 5.4% for maize at the V14 stage. However, the SPAD value was 7.9% and 5.8% with NTSM and NTSS greater than that with CT and was 5.6% with NTSM greater than that with CTS at the VT stage. NTSM, NTSS, and CTS had greater SPAD by 9.7%, 8.5%, and 7.2% at the R2 stage and greater by 15.9%, 14.1%, and 7.6% at the R3 stage than CT, but no significant difference was not found among the three wheat straw-returning treatments. At the R4 stage, NTSM and NTSS had greater SPAD by 17.5% and 15.1% than CT and 10.6% and 8.4% than CTS, respectively. Although the SPAD value of maize was decreased with NTSM and NTSS at V8 and V14 stages, a higher SPAD value with NTSM and NTSS was maintained at R2, R3, and R4 stages across the reproductive growth period, and the SPAD value with NTSM was greater than that with NTSS, it indicated that NTSM treatment can maintain high physiological activity for maize during the late growth period.

#### 2.2.2. Net Photosynthetic Rate

The net photosynthetic rate (Pn) of green leaves for maize was significantly affected by wheat straw-returning approaches (Figure 4). NTSM and NTSS decreased the Pn of green leaves for maize by 5.2% and 6.0% compared to CT at the V8 stage. However, at the VT stage, NTSM and NTSS increased the Pn of green leaves for maize by 7.7% and 6.6% compared to CT. NTSM, NTSS, and CTS had greater Pn by 15.2%, 10.2%, and 9.4% at the R2 stage and greater by 19.2%, 11.0%, and 11.2% at the R3 stage than CT and NTSM had greater Pn by 5.2% at R2 stage and 7.2% at R3 stage than CTS. At the R4 stage, NTSM and NTSS had greater Pn by 17.1% and 15.6% than CT and 7.7% and 6.3% than CTS, respectively. NTSM and NTSS maintained a higher Pn across the reproductive growth period, which lays the foundation for a high yield of maize.

#### 2.2.3. Transpiration Rate

Wheat straw-returning approaches had a significant effect on the transpiration rate (Tr) of plastic film-mulched maize from the VT to the R4 stage, except for the V8 and V14 stages (Figure 5). At the VT stage, only NTSM had greater Tr by 5.5% than CT. Compared to CT, NTSM, NTSS, and CTS increased Tr by 8.4%, 7.7%, and 5.0% at the R2 stage, increased by 12.1%, 10.4%, and 7.8% at the R3 stage, and increased by 9.9%, 8.6%, and 5.5% at R4 stage, respectively. Although there was no difference in Tr among the three wheat straw-returning treatments, NTSM showed a trend of increasing Tr, which is conducive to improving effective water use.

#### 2.2.4. Leaf Water Use Efficiency

Wheat straw returning had no significant effect on leaf water use efficiency (WUE_L_) of maize with plastic film mulching at V8, V14, and VT stages across the vegetative period (Figure 6). At the R2 stage, only NTSM increased the WUE_L_ of maize by 6.2% in comparison to CT. In addition, the WUE_L_ value of maize with NTSM and NTSS was 6.7% and 5.9% at the R3 stage and 6.6% and 6.4% at the R4 stage than that with CT. The results indicated that the higher WUE_L_ was obtained by NTSM, and NTSS was mainly reflected at the grain-filling stage of maize. This was conducive to the efficient use of soil moisture.

### 2.3. Grain Yield of Maize Affected by Various Wheat Straw-Returning Approaches

Wheat straw returning increased the grain yield of maize with plastic film mulching by 12.1% compared to CT (Table 1). NTSM was the most productive treatment, increasing the grain yield of maize by 15.6% and 7.2% compared to CT and CTS, respectively.

Previous wheat straw return significantly affected the ear number (EN), grain number per ear (GNE), and 100-grain weight (HGW) of maize with plastic film mulching (Table 1). Compared to CT, NTSM, and NTSS treatments increased EN by 19.5% and 10.5%, increased HGW by 9.2% and 7.8%, and increased GNE by 65.8%, 62.6%, and 43.4%. NTSM had a higher increasing effect on EN, GNE, and HGW than NTSS, and NTSM higher the three indexes by 14.0%, 15.6%, and 5.3% than CT, respectively. So, the increase in grain yield of maize by NTSM was attributed to the synergic increase in yield components.

### 2.4. Relationship among Grain Yield and Photosynthetic Physiological Parameters and Yield Components of Maize

#### 2.4.1. Correlation Analysis

Principal component analysis (PCA) was conducted to explain the relationship between maize grain yield (GY) and photosynthetic physiological parameters and yield components in two studied years (Figure 7). The first principal component was composed of GY, leaf area duration (LAD), leaf area index (LAI), net photosynthesis rate (Pn), ear number (EN), chlorophyll relative content (SPAD), 100-grain weight (HGW), and grain number per ear (GNE), and its contribution rate reached 91.8%. The second principal component was composed of transpiration rate (Tr) and leaf water use efficiency (WUE_L_), and its contribution rate reached 5.5%. GY was very significantly positively correlated with LAD, LAI, SPAD, Pn, EN, and HGW, and GY was significantly positively correlated with WUE_L_, GNE, and Tr. The above analysis indicates that the GY of maize with plastic film mulching was improved by no tillage with wheat straw mulching due to the coordination of yield components through maintaining a greater photosynthetic source and improving the net photosynthesis rate of green leaves.

#### 2.4.2. Incidence Matrix Analysis

In order to clarify the influence degree of various factors on the grain yield of maize, the incidence matrix analysis of grain yield and its influencing factors showed that the dominant factor affecting the grain yield of maize is the photosynthetic source, followed by photosynthetic physiological parameters and yield components (Table 2). The order of influencing the degree of grain yield for maize was LAD, LAI, Pn, EN, HGW, SPAD, Tr, WUE_L_, and GNE. The above results showed that no tillage with wheat straw mulching could promote the coordinated development of three yield components of maize by maintaining a greater photosynthetic source, improving the net photosynthesis rate of green leaves. It is a feasible way to enhance the grain yield of maize by using appropriate agronomic measures to coordinate and synchronously improve yield components in arid conditions.

## 3. Materials and Methods

### 3.1. Study Area

A field experiment was conducted in Wuwei City from 2009 to 2012. It is in a typical temperate arid zone, with an average annual sunshine duration greater than 2945 h, an annual mean air temperature of 7.2 °C, and a frost-free period of approximately 155 days. It is only adequate for a one-season cropping pattern. The soil at the experimental area is classified as a type of desert land filled with calcareous particles. The soil classification in the experimental area was Aridisol. At the start of the experiment, in the 0–30 cm soil layer, soil contained 14.3 g kg^−1^, 1.78 mg kg^−1^, and 12.5 mg kg^−1^ of organic matter, NH_4_^+^–N, and NO_3_^−^–N, respectively, and soil pH, bulk density, and cation exchange capacity were 8.2, 1.57 g cm^−3^, and 15.3 cmol kg^−1^, respectively. This area is a typical oasis agriculture zone that relies solely on irrigation. The low water availability limits the potential of cropping extension based on conventional tillage. Maize is the main grain crop in this region, and the continuous cropping pattern has worsened the obstacles of its productivity. Because the increasing effects of straw returning are mainly manifested in the following years, the selective data obtained from the following maize seasons of 2010 and 2012 were compared. During the maize growing season, the precipitation was 94.7 mm in 2010 and 128.5 mm in 2012.

### 3.2. Experimental Design

The experimental design was mainly to integrate wheat straw returning into the spring wheat–maize rotation system to improve maize productivity. The application of four wheat straw-returning approaches to maize production under plastic film mulching constituted four treatments as follows: (i) NTSM, no tillage with wheat straw mulching (no tillage with 25 to 30 cm length wheat straw that was chopped and evenly spread on the soil surface at wheat harvest the previous season); (ii) NTSS, no tillage with wheat straw standing (no tillage with 25 to 30 cm length wheat straw standing in the field after wheat harvest the previous season); (iii) CTS, conventional tillage with wheat straw incorporation (25 to 30 cm length wheat straw was chopped and incorporated into the soil through conventional tillage with the depth 30 cm at wheat harvest the previous season); and (iv) CT, conventional tillage without wheat straw returning (the control treatment, conventional tillage with the depth 30 cm was applied to the plot with straw removed from the field). In all, there are four treatments tested using a completely randomized block design with three replicates constituting a total of 12 plots (Table 3). From late July to early August in 2009 and 2011, spring wheat treatments were managed as described above at the harvest stage. In the next spring (2010 and 2012), first, fertilizing, rotary tillage, harrowing, smoothing, and compacting at the previous wheat fields; then, plastic film mulching on the wheat straw surface, and maize planted by dibbler. Spring wheat treatments were rotated with maize in alternate years (Table 3). This was performed to provide a spring wheat–maize rotation system to avoid potential weaknesses of continuous cultivation.

Spring wheat (cultivar Yong-liang 4) was planted on 29 March 2009 and 28 March 2011 and harvested on 24 July 2009 and 22 July 2011, respectively. Maize (cultivar Wu-ke 2) was planted on 22 April 2010 and 20 April 2012 and harvested on 28 September 2010 and 2 October 2012, respectively. Each plot was 48 m^2^. The planting density was 82,500 plants ha^−1^; urea (46-0-0 of N-P_2_O_5_-K_2_O) and diammonium phosphate (18-46-0 of N-P_2_O_5_-K_2_O) was broadcast and incorporated into the soil at sowing. In this study, all treatments received 450 kg N ha^−1^ and 225 kg P_2_O_5_ ha^−1^. All P was applied as base fertilizer, while 30% N was applied at sowing, 60% top-dressed at jointing, and the remaining 10% top-dressed at grain filling. For the top-dressed N applications for maize, a 3 cm diameter hole (10 cm deep) was made 4–5 cm away from the maize seeding hole, fertilizer was applied into the hole, and the hole was compacted with soil.

The crops were irrigated according to the recommendations for optimizing crop production in the local area due to low precipitation. All of the plots received 1200 m^3^ ha^−1^ of irrigation in late fall just before soil freezing, and then various irrigation quotas were applied at different maize growth stages, such as 900, 750, 900, 750, and 750 m^3^ ha^−1^ of irrigation water were applied at the V4, V8, V14, VT, and R2 stages of maize. A hydrant pipe system was used for irrigation, and a flow meter was installed at the discharging end of the pipe to record the irrigation volume entering each plot.

### 3.3. Experimental Data Collection

#### 3.3.1. Leaf Area Index and Leaf Area Duration

After the emergence of maize, five maize plants were selected randomly by S-shape to measure the leaf area in each plot at various growth stages (Table 4). The leaf length (*ai*) and the greatest leaf width (*bi*) were measured using a ruler, and the leaf area index (LAI) and leaf area duration (LAD) was determined using the following formula:(1)$$\mathrm{LAI}=0.75\times P\times \sum_{i=1}^{n}\left(ai\times bi\right)$$
(2)$$\mathrm{LAD}=\sum_{i=1}^{n}\left(LAIi\times Di\right)$$
where *P* is the seedling number of maize, 0.75 is the compensation coefficient of maize, and *Di* is the duration of one growth stage.

#### 3.3.2. Chlorophyll Relative Content

The chlorophyll relative content (SPAD) with the three fully expanded top leaves of maize in the center of each plot was measured at V8, V14, and VT stages. After these stages, the three ear leaves were measured at R2, R3, and R4 stages.

#### 3.3.3. Photosynthetic Indices

In this study, net photosynthesis rate (Pn) and transpiration rate (Tr) with the three fully expanded top leaves of maize in the center of each plot were measured by a Portable Photosynthesis System (LI-Cor 6400XT, Lincoln, NE, USA) at V8, V14, and VT stages. After these stages, the three ear leaves were measured at R2, R3, and R4 stages. At the same time, leaf water use efficiency (WUE_L_) was determined by Pn divided by Tr.

#### 3.3.4. Grain Yield and Its Components

The grain yield was determined by using a small combine harvester at the physiological maturity stage of maize. The sampling square of 5 m was selected to investigate the ear number (EN) for maize in each plot. In addition, 10 maize plants were randomly sampled to determine the grain number per ear (GNE). The grain yield per unit area and 100-grain weight (HGW) are converted to the standard grain water content of 13%.

### 3.4. Statistical Analysis

All statistical analyses were performed using SPSS version 20.0 (SPSS Inc., Chicago, IL, USA). The least significant difference test was used to conduct multiple comparisons at *p* < 0.05. The relationships between the grain yield and photosynthetic physiological parameters and yield components were determined using principal component analysis (PCA) and incidence matrix analysis.

## 4. Discussion

### 4.1. Effects of Straw Returning on Photosynthetic Sources of Plastic Film-Mulched Maize

Leaves are the most important component of the photosynthetic source of crops, and increasing green leaf area and extending green holding time play a key role in light energy utilization, soil water and nutrient absorption, and yield formation [30]. In this study, wheat straw-returning treatments increased the leaf area index (LAI) and leaf area duration (LAD) of maize, and no tillage with straw mulching (NTSM) had the greater increasing effect on LAI and LAD of maize with plastic film mulching. This resulted in a solid foundation for a high yield of maize. Meanwhile, no tillage in combination with wheat straw returning decreased the LAI and LAD of rotated maize before the VT stage but increased the LAI and LAD after the VT stage, which effectively regulated the growth and development of plastic film-mulched maize during the early and late stages of development. In particular, NTSM had a better effect on regulating photosynthetic sources than other treatments. This resulted from no tillage with wheat straw mulching in favor of root growth under conditions of low temperature during the early stage of maize development and delayed the aboveground growth of maize, leading to a decreased photosynthetic source [31]. A traditional plastic film without wheat straw returns accelerated plant senescence under extremely high soil temperatures (>35 °C in the 0–25 cm soil layer) from the VT to R4 stage of maize [28,32]. In addition, the excessive consumption of water and nutrients during the early stage of maize development caused nutrient deficiency during the later stages, leading to a small photosynthetic source for the traditional plastic film without straw returning [33]. In addition, no tillage with wheat straw returning had an appropriate soil temperature at the late stage of maize development and delayed root and leaf senescence, thus maintaining the larger photosynthetic source of plastic film-mulched maize [28,34]. Since nutrient uptake depends on the continuous carbohydrate supply from the shoot to the roots, the longer photosynthetic time in the reproductive growth stage is conducive to the absorption of plant nutrients [35], which in turn increases the canopy photosynthesis duration. The surplus nutrients in the early growth period of maize are used for the vigorous growth demand in the later period [33], thus improving the photosynthetic source of maize with plastic film mulching after the VT stage. The above analyses fully show that no tillage with wheat straw returning could enhance the yield improvement potential by maintaining a higher LAI and LAD after the VT stage for plastic film-mulched maize.

### 4.2. Effects of Straw Returning on Physiological Characteristics of Plastic Film-Mulched Maize

In this study, no tillage with wheat straw mulching/standing (NTSM and NTSS) decreased SPAD, Pn, and Tr of green leaves before the VT stage of plastic film-mulched maize, which was because no tillage with wheat straw returning decreased the growth rate and consumed less moisture and nutrients of maize at low air temperature season [36,37], thus reducing the photosynthetic capacity of green leaves of plastic film-mulched maize. On the contrary, as soil temperature reached the suitable heat demand of maize with the rise of air temperature [28,36], the remaining soil moisture and nutrients before the VT stage boosted the vigorous growth of maize after the VT stage [34]. Published studies have shown that no tillage with straw returning inhibited soil evaporation, increased water infiltration, and maintained relatively suitable soil moisture conditions [19,38], thus enhancing the photosynthetic capacity of green leaves. In addition, a relatively moist soil water environment and suitable water conditions for plant leaves can improve the relative chlorophyll content (SPAD), net photosynthetic rate (Pn), and transpiration rate (Tr) of green leaves, especially in the late growing period of crops [39,40]. So, the values on SPAD, Pn, and Tr of green leaves in maize with NTSM and NTSS were greater than CT at the reproductive period (R2, R3, and R3 stages), and NTSM had a higher increasing effect than NTSS. The main reasons were that NTSM was most effective in conserving soil moisture during the entire maize growth period, the soil moisture loss was slow, and the available moisture was kept for a long time, compared to NTSS [27,36,41]. Meanwhile, NTSM effectively regulated the dynamics of photosynthetic sources of green leaves for the maize growth period, guaranteeing a large LAI at the later growth period of maize, so the values of SPAD and Pn were higher than other treatments.

### 4.3. Grain Yield of Plastic Film-Mulched Maize with Was Affected by Straw Returning

In the present study, wheat straw returning was integrated into maize production with plastic film mulching and significantly promoted maize yield increase. Conventional without wheat straw returning had lower grain yield of plastic film-mulched maize than wheat straw returning. This was because single plastic film mulching could be detrimental to crop yield improvement and stability under extremely high soil temperatures because it accelerates root senescence of plants [29,32], thus reducing the potential for grain increase. Obviously, no tillage with wheat straw returning boosted the grain yield of plastic film-mulched maize via the synergic increase in ear number (EN), grain number per ear (GNE), and 100-grain weight (HGW). Across the wheat and straw-returning treatments, the grain yield increase in NTSM was greater than that of NTSS and CTS.

The high yield of NTSM comes from several aspects via the analysis of photosynthetic source size and photosynthetic physiology traits. First, NTSM effectively regulated the dynamics of the photosynthetic source of green leaves for the maize growth period in this study, guaranteeing a higher leaf area index in the later growth period, which is beneficial to maintain higher photo-physiological characteristics and promote grain filling for achieving high yield. Second, the values of SPAD, Pn, and Tr of green leaves in maize at the R2, R3, and R3 stages were higher under NTSM than in other treatments. It showed that NTSM could maintain the higher physiological activity of green leaves in maize across the late growth period. Third, the growth period and functional period of green leaves of maize were prolonged by NTSM [37], thus maintaining a higher photosynthetic capacity of green leaves in maize. Additionally, many publications have indicated that no tillage with straw returning enhanced the distribution and translocation of photoassimilates from vegetative organs to reproductive organs compared to conventional tillage without straw returning [42,43]. A similar finding is reflected in this study. NTSM promoted grain filling and increased 100-grain weight. Therefore, no tillage with wheat straw mulching can not only meet the increasing population’s demand for food but also alleviate the shortage of water resources in arid regions.

## 5. Conclusions

No tillage with wheat straw mulching (NTSM) effectively regulated the dynamics of a photosynthetic source of maize with plastic film mulching across the growth period, guaranteeing a large leaf area index and duration at the later growth period of maize. Compared to CT control, NTSM increased relative chlorophyll content, net photosynthetic rate, transpiration rate, and leaf water use efficiency of green leaves in maize plants from VT to R4 stage, which can keep a higher physiological activity at the late growth period of maize. Thereby, NTSM boosted the grain yield of maize by 15.6% in comparison to CT, and the high yield was attributed to the synchronous increase and cooperative development of ear number, grain number per ear, and 100-grain weight. This study concluded that no tillage with straw mulching could be recommended as a promising previous straw management technique for maize production with plastic film mulching to coordinate the conflict between high yield and water scarcity in arid conditions.

## Figures and Tables

**Figure 1 plants-12-01358-f001:**
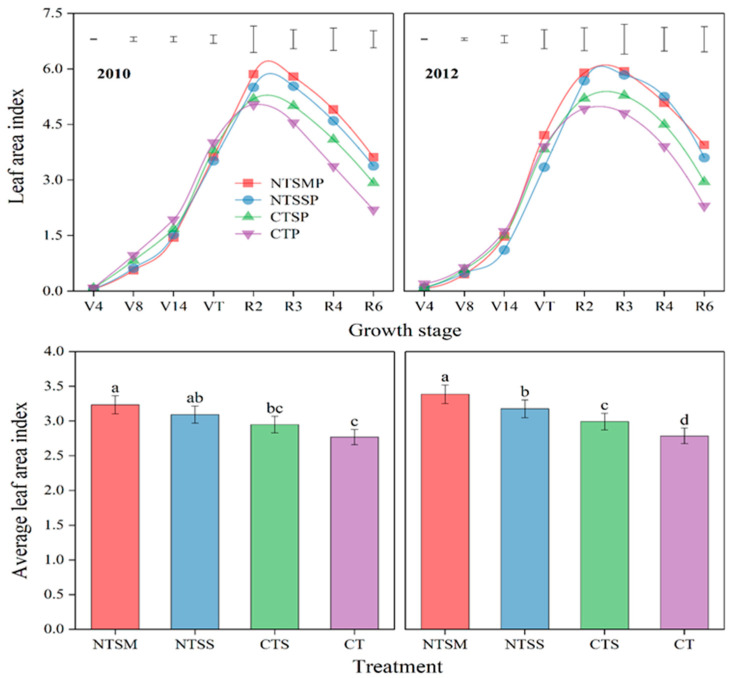
Response on dynamics of leaf area index and average leaf area index of maize with plastic film mulching to wheat straw-returning approaches. The length of vertical bars represents the magnitude of the least significant difference (LSD) at *p* = 0.05 among treatments within a growth stage. Different lowercase letters indicate treatment means that are significantly different at *p* < 0.05. Treatment abbreviations are described in Section 2.3.

**Figure 2 plants-12-01358-f002:**
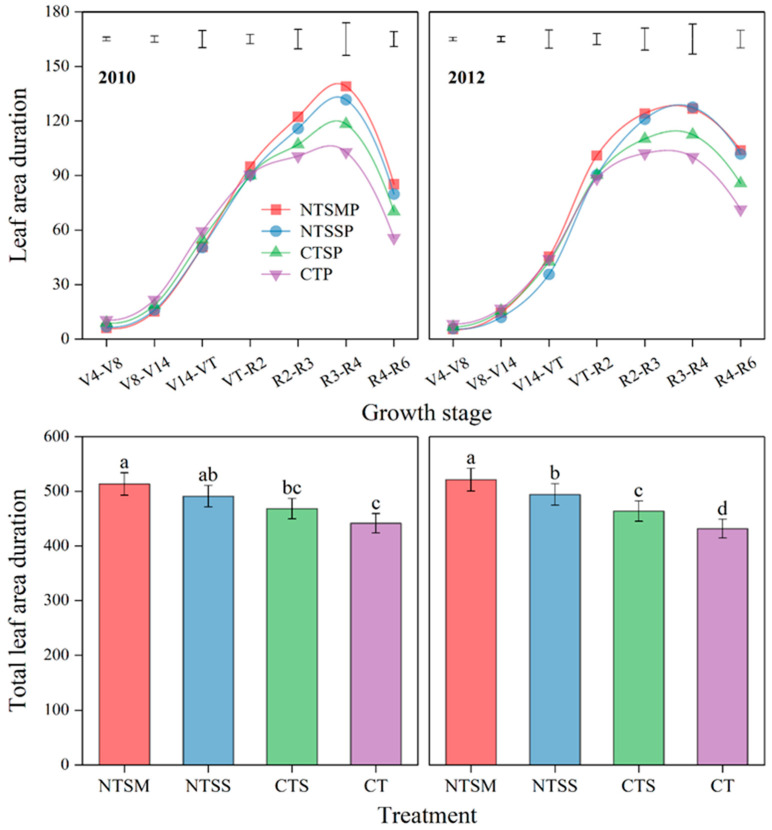
Response on dynamics of leaf area duration and total leaf area duration of maize with plastic film mulching to wheat straw-returning approaches. The length of vertical bars represents the magnitude of the least significant difference (LSD) at *p* = 0.05 among treatments within a growth stage. Different lowercase letters indicate treatment means that are significantly different at *p* < 0.05. Treatment abbreviations are described in Section 2.3.

**Figure 3 plants-12-01358-f003:**
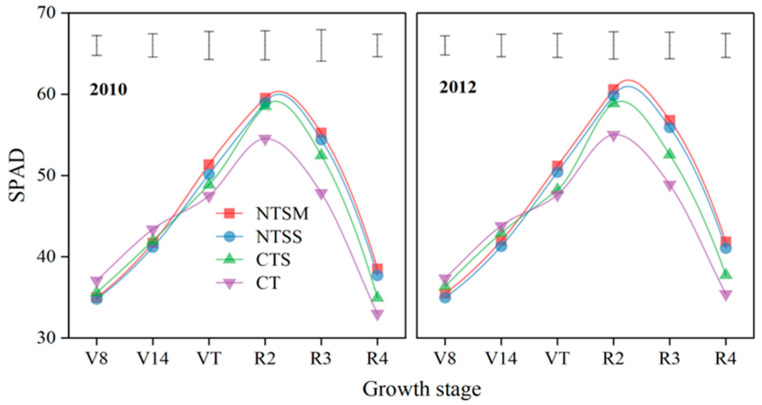
Dynamics of chlorophyll relative content (SPAD) for green leaves in maize with plastic film mulching as affected by various wheat straw-returning approaches. The length of vertical bars represents the magnitude of the least significant difference (LSD) at *p* = 0.05 among treatments within a growth stage. Treatment abbreviations are described in Section 2.3.

**Figure 4 plants-12-01358-f004:**
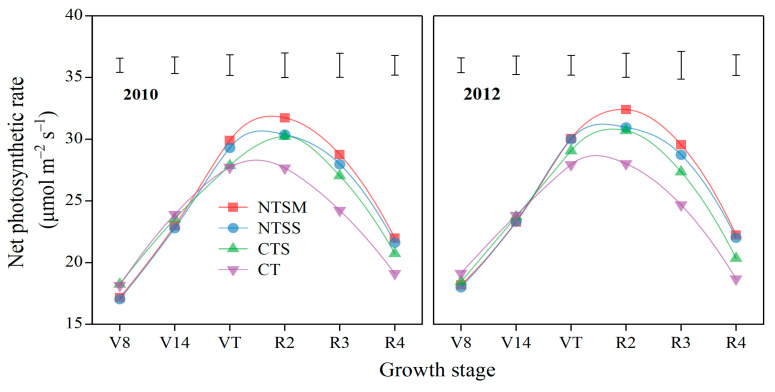
Dynamics of net photosynthetic rate (Pn) for green leaves in maize with plastic film mulching as affected by various wheat straw-returning approaches. The length of vertical bars represents the magnitude of the least significant difference (LSD) at *p* = 0.05 among treatments within a growth stage. Treatment abbreviations are described in Section 2.3.

**Figure 5 plants-12-01358-f005:**
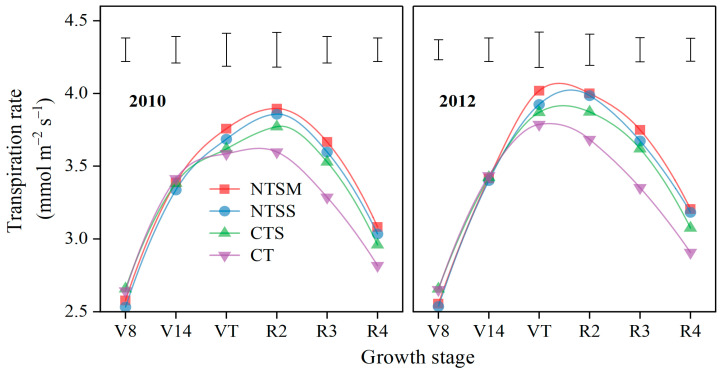
Dynamics of transpiration rate (Tr) for green leaves in maize with plastic film mulching as affected by various wheat straw-returning approaches. The length of vertical bars represents the magnitude of the least significant difference (LSD) at *p* = 0.05 among treatments within a growth stage. Treatment abbreviations are described in Section 2.3.

**Figure 6 plants-12-01358-f006:**
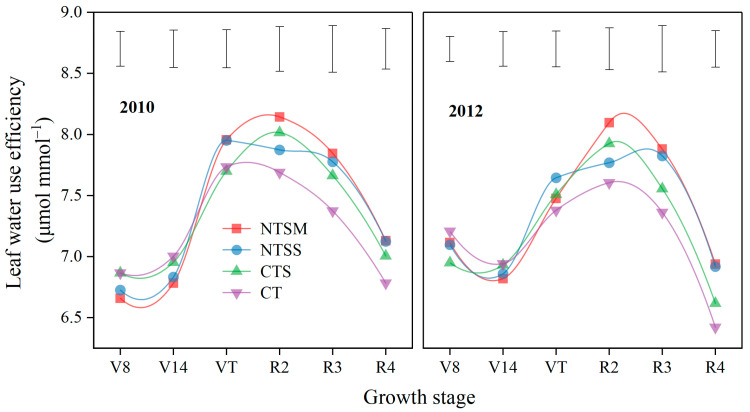
Dynamics of leaf water use efficiency (WUE_L_) in maize with plastic film mulching as affected by various wheat straw-returning approaches. The length of vertical bars represents the magnitude of the least significant difference (LSD) at *p* = 0.05 among treatments within a growth stage. Treatment abbreviations are described in Section 2.3.

**Figure 7 plants-12-01358-f007:**
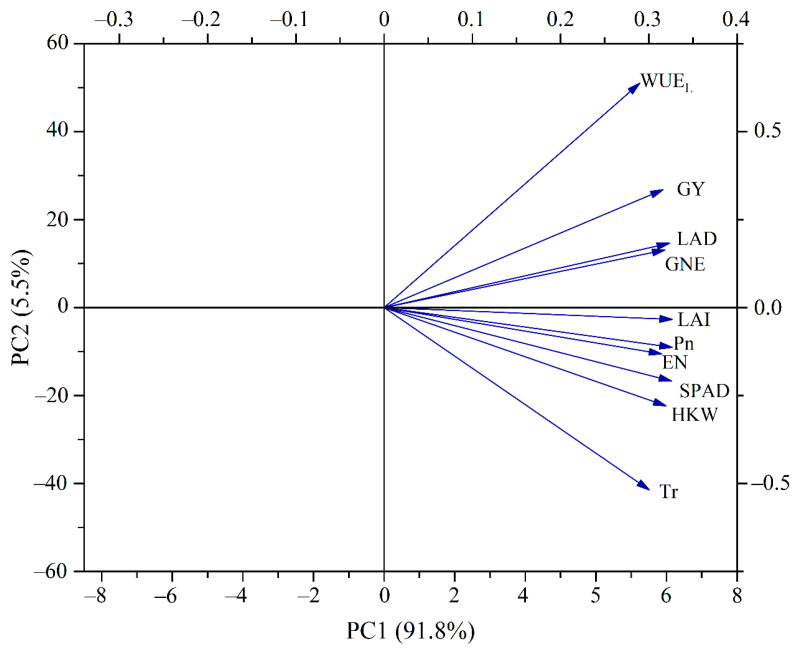
Principal component analysis (PCA) was conducted based on the comprehensive data of grain yield (GY) and photosynthetic physiological parameters and yield components in 2010 and 2012.

**Table 1 plants-12-01358-t001:** Grain yield and its components of maize with plastic film mulching as affected by various wheat straw-returning approaches.

Year	Treatment ^†^	Grain Yield (t ha^−1^)	Yield Component		
			Ear Number Per Area (Ear m^–2^)	Grain Number Per Ear (Grain Ear^−1^)	100-Grain Weight (g)
2010	NTSM	13.47 a ^‡^	8.50 a	544.8 a	34.93 a
	NTSS	13.05 ab	7.93 b	534.5 a	34.35 a
	CTS	12.76 b	7.42 c	469.2 b	33.09 ab
	CT	11.46 c	7.03 d	332.8 c	31.86 b
2012	NTSM	13.25 a	8.73 a	553.0 a	35.84 a
	NTSS	13.05 a	8.04 ab	542.4 a	35.35 a
	CTS	12.16 b	7.71 b	480.7 b	34.13 ab
	CT	11.65 c	7.39 b	329.6 c	32.96 b

^†^ The descriptions of the treatment names were defined in Table 1. ^‡^ Within a column for a given year, means followed by different letters are significantly different at *p* < 0.05.

**Table 2 plants-12-01358-t002:** The incidence matrix among grain yield and their impact factors as well as ranking for maize at arid irrigated regions.

Degree of Association	Photosynthetic Source		Photosynthetic Physiological Parameters				Yield Components		
	LAI	LAD	SPAD	Pn	Tr	WUE_L_	EN	GNE	HGW
Grain yield	0.8640	0.8884	0.7953	0.8193	0.7607	0.7591	0.8192	0.5471	0.8161
Ranking	2	1	6	3	7	8	4	9	5

LAI, leaf area index; LAD, leaf area duration; SPAD, chlorophyll relative content; Pn, net photosynthesis rate; Tr, transpiration rate; WUE_L_, leaf water use efficiency; EN, ear number; GNE, grain number per ear; HGW, 100-grain weight.

**Table 3 plants-12-01358-t003:** The detailed description of treatments.

Treatment Code	Previous Wheat Straw Management for the Following Maize Production	Crop Types in Different Years 2009–2010–2011–2012
NTSM	No tillage with wheat straw mulching for the length of 25–30 cm	Spring wheat–maize–spring wheat–maize
NTSS	No tillage with wheat straw standing for the length of 25–30 cm	
CTS	Conventional tillage with wheat straw incorporation for the length of 25–30 cm	
CT	Conventional tillage without wheat straw returning	

The experiment in 2009 and 2011 was performed to provide various wheat straw-returning approaches for the treatments that were implemented in the following years in 2010 and 2012 for maize with plastic film mulching.

**Table 4 plants-12-01358-t004:** The growth stage of maize at sampling times.

Sampling Order	The Growth Stage of Maize in Sampling	
	Chinese Name	International Name
First	Emergence stage	VE
Second	Fourth-leaf stage	V4
Third	Eighth-leaf stage	V8
Fourth	Fourteenth-leaf stage	V14
Fifth	Tasseling stage	VT
Sixth	Blister kernel stage	R2
Seventh	Milking stage	R3
Eighteen	Doughing stage	R4
Ninth	Maturing stage	R6

## Data Availability

The entire set of raw data presented in this study is available upon request from the corresponding author.

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
