# Peer review of "Photosynthetic Physiological Basis of No Tillage with Wheat Straw Returning to Improve Maize Yield with Plastic Film Mulching in Arid Irrigated Areas"

_plants, 2023, doi:10.3390/plants12061358_

Round 1

Reviewer 1 Report

This manuscript is a good start to better understanding the crop physiological and yield impacts of no-till and tillage in wheat preceding corn mulched with plastic. However, the co-authors need to make improvements to the manuscript in order for this to be suitable for publication in MDPI Plants. I would be willing to continue to review this manuscript for continuous improvement after the authors have made the following TWELVE substantive edits as well as minor line-number specific edits:

    1)      There is no space between double citations so please correct to “[#,#]” throughout the manuscript. 

    2)      The abbreviations used for the cropping systems all end in the letter “P” so please delete this since all systems have corn mulched with plastic in common. It will reduce the length of these abbreviations which will make the manuscript more readable. So please change these to: NTSM, NTSS, CTS, and CT.

    3)      Please avoid the use of abbreviations in the abstract to the extent that they are used. Verbally describe what you are saying better so you do not have to use these as much. 

    4)      For all figures, please add x-axis labels. 

    5)      For Table 1 and any tables and figures were this is not done, please add a blank row both above and below the table or figure.

    6)      For each major section of the manuscript, when using abbreviations please follow this guideline of writing out and first, writing out and having the abbreviation in parentheses, and then using the abbreviation after this if it is repeated used. 

    7)      Earlier in the manuscript, please delete any phrase such as “As we know…” or “As is known to all…” which you will find on L37 and L57 respectively. Not everyone reading your manuscript knows these things. If you are stating that this is known to all, that is impossible since for something this discipline specific, there is no way that everyone would know whatever it is that you are writing. 

    8)      In the Results section, all the writing has an overuse of abbreviations and listing of percent ranges. The writing is very repetitive which makes is very challenging for the reader to want to read what is being written. Please correct this and consult with a professional English editor if needed. 

    9)      It is not clear from the methods section if the wheat planted is winter wheat or spring wheat since the dates of plant and harvest are not stated. It is not stated when the corn was harvested.

    10)  The titles for sub-sections especially in the Discussion section are too long and not clear. 

    11)  In the Discussion section you need to summarize more clearly what may be causing the results particularly as they pertain to the soil ecology in no-till systems versus tilled systems and cite journal articles on this. What is it about tilling that results in such statistically significant yield declines in corn mulched with plastic? It is also worth briefly mentioning that carbon residue can increase nitrogen immobilization and that there was sufficient application of fertilizers (particularly nitrogen) to counteract such nutrient immobilization. 

    12)  The volume(issue) number in the references needs to be in italics. While some journals only use volume numbers, many also have issue numbers and these need to be included. You also need to include the DOI link for cross referencing after the page numbers.

Specific Line Number of Manuscript Figure/Table comments (note that requested change of word(s) in quotations where NO edits need to be made for writing before/after each “…”):

L98 – 2985°C?

L9-14 – Change to: 155 days

Author Response

Comments and Suggestions for Authors

This manuscript is a good start to better understanding the crop physiological and yield impacts of no-till and tillage in wheat preceding corn mulched with plastic. However, the co-authors need to make improvements to the manuscript in order for this to be suitable for publication in MDPI Plants. I would be willing to continue to review this manuscript for continuous improvement after the authors have made the following TWELVE substantive edits as well as minor line-number specific edits:

    1) There is no space between double citations so please correct to “[#,#]” throughout the manuscript.

Response: OK! These problems have been addressed.

    2) The abbreviations used for the cropping systems all end in the letter “P” so please delete this since all systems have corn mulched with plastic in common. It will reduce the length of these abbreviations which will make the manuscript more readable. So please change these to: NTSM, NTSS, CTS, and CT.

Response: Thanks for your review and valuable comments. The treatment codes have been modified as: NTSM, NTSS, CTS, and CT.

    3) Please avoid the use of abbreviations in the abstract to the extent that they are used. Verbally describe what you are saying better so you do not have to use these as much.

Response: Very good comments. The abbreviations in the abstract have been revised.

4) For all figures, please add x-axis labels.

Response: Very detailed review. The x-axis labels have been added in the all figures.

    5) For Table 1 and any tables and figures were this is not done, please add a blank row both above and below the table or figure.

Response: Very detailed review. A blank row was added of both above and below the table or figure in the text.

6) For each major section of the manuscript, when using abbreviations please follow this guideline of writing out and first, writing out and having the abbreviation in parentheses, and then using the abbreviation after this if it is repeated used.

Response: Very good advice. Such proposals have been revised in detail in the text.

7) Earlier in the manuscript, please delete any phrase such as “As we know…” or “As is known to all…” which you will find on L37 and L57 respectively. Not everyone reading your manuscript knows these things. If you are stating that this is known to all, that is impossible since for something this discipline specific, there is no way that everyone would know whatever it is that you are writing.

Response: Good comment. Such proposals have been revised in detail in the text. The phrase such as “As we know…” or “As is known to all…” have been deleted.

8) In the Results section, all the writing has an overuse of abbreviations and listing of percent ranges. The writing is very repetitive which makes is very challenging for the reader to want to read what is being written. Please correct this and consult with a professional English editor if needed.

Response: Very good advice. All authors have made detailed revisions to the results analysis, reducing the use of abbreviations and percentage ranges.

9) It is not clear from the methods section if the wheat planted is winter wheat or spring wheat since the dates of plant and harvest are not stated. It is not stated when the corn was harvested.

Response: Very detailed review. In this study, the wheat planted is spring wheat. Spring wheat was planted on 29 March in 2009 and 28 March in 2011, and harvested on 24 July in 2009 and 22 July in 2011, respectively. Maize was planted on 22 April in 2010 and 20 April in 2012, and harvested on 28 September in 2010 and 2 October in 2012, respectively.

    10) The titles for sub-sections especially in the Discussion section are too long and not clear.

Response: Very good comments. The titles for sub-sections especially in the Discussion section have been revised.

11) In the Discussion section you need to summarize more clearly what may be causing the results particularly as they pertain to the soil ecology in no-till systems versus tilled systems and cite journal articles on this. What is it about tilling those results in such statistically significant yield declines in corn mulched with plastic? It is also worth briefly mentioning that carbon residue can increase nitrogen immobilization and that there was sufficient application of fertilizers (particularly nitrogen) to counteract such nutrient immobilization.

Response: Very good comments. All authors have made detailed revisions to the discussion and strengthened the mechanism analysis.

12) The volume(issue) number in the references needs to be in italics. While some journals only use volume numbers, many also have issue numbers and these need to be included. You also need to include the DOI link for cross referencing after the page numbers.

Response: Very good comments. The form of the references has been revised in detail.

Specific Line Number of Manuscript Figure/Table comments (note that requested change of word(s) in quotations where NO edits need to be made for writing before/after each “…”):

Response: OK. The above problems have been addressed.

L98 – 2985°C?

Response: It is correct. accumulated air temperature more than 10°C higher than 2985°C.

L9-14 – Change to: 155 days

Response: OK. It has been changed to 155 days.

Reviewer 2 Report

The manuscript entitled “Photosynthetic physiological basis of no-tillage with wheat straw returning to improve maize yield with plastic film mulching in arid irrigated areas” by Guo et al aimed to study how plastic film combined with different wheat straw returning approaches to affect the maize yield. They found that one of four approaches, no-tillage with wheat straw mulching (NTSMP) has positive effects on regulating the photosynthetic physiological traits, and a higher grain yield of maize. This study provides insights into the improvement of plant productions in arid and water-scarce areas. 

Overall, the method used in the study is thorough. The authors provided the data to support the conclusions. Statistical analysis is provided within the manuscript. There are some minor concerns before I recommend accepting it:

1. It is not appropriate to present the range of data in the manuscript. 

For example, Line 186 On average, the LAI of three wheat straw returning treatments was 17.8 - 30.7%, 26.2 - 31.1%, and 11.2%- 13.6% lower than that of conventional tillage without wheat straw returning (CTP).  The authors should take the appropriate way to present the data in the whole manuscript. I suggest showing the average value instead of the range of change (increase/decrease). 

2. Line 319- 323, “LAD (leaf area duration) ” should be “leaf area duration (LAD)”. Similarly, the authors should proofread the manuscript. 

3. For figure 7, it is not clear for the data that the authors used to do PCA analysis. Is it from 2010 or 2012? Please describe the details in the method. 

Author Response

The manuscript entitled “Photosynthetic physiological basis of no-tillage with wheat straw returning to improve maize yield with plastic film mulching in arid irrigated areas” by Guo et al aimed to study how plastic film combined with different wheat straw returning approaches to affect the maize yield. They found that one of four approaches, no-tillage with wheat straw mulching (NTSMP) has positive effects on regulating the photosynthetic physiological traits, and a higher grain yield of maize. This study provides insights into the improvement of plant productions in arid and water-scarce areas.

Overall, the method used in the study is thorough. The authors provided the data to support the conclusions. Statistical analysis is provided within the manuscript. There are some minor concerns before I recommend accepting it:

  1. It is not appropriate to present the range of data in the manuscript. For example, Line 186 On average, the LAI of three wheat straw returning treatments was 17.8 - 30.7%, 26.2 - 31.1%, and 11.2%- 13.6% lower than that of conventional tillage without wheat straw returning (CTP). The authors should take the appropriate way to present the data in the whole manuscript. I suggest showing the average value instead of the range of change (increase/decrease).

Response: Very good comment. It has been revised in the text, combined with the comment of reviewer #1.

  1. Line 319- 323, “LAD (leaf area duration)” should be “leaf area duration (LAD)”. Similarly, the authors should proofread the manuscript.

Response: Good advice. It has been revised in the text.

  1. For figure 7, it is not clear for the data that the authors used to do PCA analysis. Is it from 2010 or 2012? Please describe the details in the method.

Response: Very detailed review. PCA analysis was conducted based on the comprehensive data of 2010 and 2012.

Reviewer 3 Report

Comments

I find the present researche work of of interest in applied science. The paper is well presented with a proper structure. conclussions are well based on the obtained results.

L. 87. I believe that the statement "optimizing photosynthetic physiological characteristics would increase crop production" it is not a real hypothesis; it is something rather obvious.

L. 99-100. I miss some soil data as pH, EC, Organic matter content, Texture, Cation exchange capacity or CaCO3 content.

L. 123. Treatments should be describe clearer both in the text and in Table 1. Such as I have understood the soil at the four treatments have plastic mulching. There are two treatmens with conventionl tillage (CTSP and CTP), one with wheat straw and the other without it. And the two other treatments with no tillage system, one with chopped wheat straw (NTSMP), and the other treatment (NTSSP), with standing dry wheat plants. Is it correct?. I am not sure how is the straw at CTSP treatment: is it chopped?. 

L. 128-129. “Soil nutrient required” …… or .. “weakness of continous cultivation”.

Plant demand of nutrients only depends on the fertilizers applied. This in addition to the important amount of water added (5,250 m3/ha) strongly remind to a hydroponic culture, where soil does only acts as a plant support.  

How do you add fertilizers to plants under plastic mulching?. 

L. 202. Small mistake at Figure 1: one of the two years should be 2012.

L.313. Significant numbers in Table 3 should be 3 or 4, not 5.

L. 371. Is it there any temperature measurement of the soil of the different treatments?.

L. 372. It i said “the excessive consumption of water and nutrients during the early stage of maize development”, but the experimental data of the actual water or nutrient content of the soil nor of the plant are not shown.

Comments on water and nutrient consumption are mentioned several times along the text whithout experimental data.

I think that part of the discussion is a bit artificial. It is not a surprise that the treatment with the best physiological characteristics had the best production and yield; the contrary would be difficult to justified. You find that the mixture of a no tillage system, with added chopped wheat straw under plastic mulching, provide the better physiological characteristic (measured) to obtain the higher maize yield (measured).

The NTSMP treatment would affect the water and nutrients availability at the root system, providing the optimal temperature and humidity in the soil. This is very logical but there are not experimental data in the present paper. Soil temperature or available  nutrients are not quantified.

Author Response

Comments and Suggestions for Authors

Comments

I find the present research work of interest in applied science. The paper is well presented with a proper structure. conclusions are well based on the obtained results.

  1. L. 87. I believe that the statement "optimizing photosynthetic physiological characteristics would increase crop production" it is not a real hypothesis; it is something rather obvious.

Response: Good comment. The hypothesis has been revised as: Our hypothesized that previous wheat straw returning to the field could improve crop yield by optimizing photosynthetic physiological characteristics and further coordinating yield composition for maize.

  1. L. 99-100. I miss some soil data as pH, EC, Organic matter content, Texture, Cation exchange capacity or CaCO3 content.

Response: Very good advice. Relevant information has been added in the study area. Such as: The soil classification in the experimental area was Aridisol. The soil at the Research Station is classified as a type of desert land filled with calcareous particles. At the start of the experiment, in the 0-30 cm soil layer, soil contained 14.3 g kg–1, 1.78 mg kg–1, and 12.5 mg kg–1, of organic matter, NH4+–N, and NO3–N, respectively, and soil pH, bulk density, and cation exchange capacity were 8.2, 1.57 g cm–3, and 15.3 cmol kg–1, respectively.

  1. L. 123. Treatments should be described clearer both in the text and in Table 1. Such as I have understood the soil at the four treatments have plastic mulching. There are two treatments with conventional tillage (CTSP and CTP), one with wheat straw and the other without it. And the two other treatments with no tillage system, one with chopped wheat straw (NTSMP), and the other treatment (NTSSP), with standing dry wheat plants. Is it correct?. I am not sure how is the straw at CTSP treatment: is it chopped?.

Response: Your understanding is correct; the author has made detailed changes to the treatment design in the experiment.

  1. L. 128-129. “Soil nutrient required” …… or .. “weakness of continuous cultivation”.

Response: OK. This sentence has been revised as: This was done to provide a wheat–maize rotation system to avoid potential weaknesses of continuous cultivation.

  1. Plant demand of nutrients only depends on the fertilizers applied. This in addition to the important amount of water added (5,250 m3/ha) strongly remind to a hydroponic culture, where soil does only acts as a plant support.

Response: Good question. During the maize growing season (April–September), the precipitation was 94.7 mm in 2010 and 128.5 mm in 2012. This area is a typical oasis agriculture zone that relies solely on irrigation. All of the plots received 1200 m3 ha–1 of irrigation in late fall just before soil freezing, and then various irrigation quotas were applied at different maize growth stages, such as 900, 750, 900, 750, and 750 m3 ha–1 of irrigation water were applied at the V4, V8, V14, VT, and R2 stages of maize. The irrigation amount of maize during the growth period was 4050 m3 ha–1.  

  1. How do you add fertilizers to plants under plastic mulching?.

Response: Good question. For the top-dressed N applications for maize, a 3-cm diameter hole (10-cm deep) was made 4–5 cm away from the maize seeding-hole, and fertilizer was applied into the hole, and the hole was compacted with soil. It has been added in the text.

  1. L. 202. Small mistake at Figure 1: one of the two years should be 2012.

Response: Very detailed review. It has been revised as 2012.

  1. L.313. Significant numbers in Table 3 should be 3 or 4, not 5.

Response: Good advice. Significant numbers for grain number per ear in Table 3 were revised as 4.

  1. L. 371. Is it there any temperature measurement of the soil of the different treatments?.

Response: Good comment. Traditional plastic film without wheat straw returning could be detrimental to crop yield improvement because it accelerates plant senescence under extremely high soil temperatures, from the VT to R4 stage of maize. In this study, it was CT control.

  1. L. 372. It i said “the excessive consumption of water and nutrients during the early stage of maize development”, but the experimental data of the actual water or nutrient content of the soil nor of the plant are not shown.

Response: Good comment. The reference has been added to support this view.

  1. Comments on water and nutrient consumption are mentioned several times along the text without experimental data.

Response: Good comment. The authors have added references to support the relevant arguments.

  1. I think that part of the discussion is a bit artificial. It is not a surprise that the treatment with the best physiological characteristics had the best production and yield; the contrary would be difficult to justified. You find that the mixture of a no tillage system, with added chopped wheat straw under plastic mulching, provide the better physiological characteristic (measured) to obtain the higher maize yield (measured).

Response: Good comment. The authors have made a thorough revision to the discussion and added relevant references to support the explanation of relevant reasons.

  1. The NTSMP treatment would affect the water and nutrients availability at the root system, providing the optimal temperature and humidity in the soil. This is very logical but there are not experimental data in the present paper. Soil temperature or available nutrients are not quantified.

Response: Good comment. The authors have made a thorough revision to the discussion and added relevant references to support the explanation of relevant reasons.

Reviewer 4 Report

The manuscript titled “Photosynthetic physiological basis of no-tillage with wheat straw returning to improve maize yield with plastic film mulching in arid irrigated areas” contains some important information.

The manuscript could be valuable work but I have some suggestions and comments to improve the manuscript. Please read the manuscript carefully to ensure that the message you intend to convey is clearly presented.

The abstract needs revision and should be substantiated with the addition of methods and a conclusion in brief. It should have proper results summarizing the core findings of the present investigation.

The introduction is somewhat good but does not provide the rationale of the study. The objectives need revision. 

The methodologies and protocols used are adequate but lack consistency. These have been described in adequate detail. The authors need to revise this section to make some of the statements meaningful and understandable.

Results although presented in detail but require some revision to clarify some of the statements.

The results have not been discussed properly. It is advisable to discuss the results with the findings of other researchers. Some of the statements are redundant.

The conclusion should be revised and give some recommendations on the basis of their findings.

The standard of English is up to the mark but it requires improvement.

Some of the references do not conform to the style of the journal.

Double-check all the citations in the text to ensure that they are listed in the references list and vice versa.

Author Response

Comments and Suggestions for Authors

The manuscript titled “Photosynthetic physiological basis of no-tillage with wheat straw returning to improve maize yield with plastic film mulching in arid irrigated areas” contains some important information.

  1. The manuscript could be valuable work but I have some suggestions and comments to improve the manuscript. Please read the manuscript carefully to ensure that the message you intend to convey is clearly presented.

Response: Good comment. The authors have made a thorough revision in all text.

  1. The abstract needs revision and should be substantiated with the addition of methods and a conclusion in brief. It should have proper results summarizing the core findings of the present investigation.

Response: Good comment. The authors have made a thorough revision of the abstract.

  1. The introduction is somewhat good but does not provide the rationale of the study. The objectives need revision.

Response: Good comment. The authors have made a thorough revision of the introduction, added the rationale of the study, and revised the objectives. Such as: The objectives were to determine (i) the responses of photosynthetic physiological characteristics of maize with plastic film mulching to previous wheat straw management practices; 2) the effects of previous wheat straw management practices on grain yield and yield components of maize with plastic film mulching.

  1. The methodologies and protocols used are adequate but lack consistency. These have been described in adequate detail. The authors need to revise this section to make some of the statements meaningful and understandable.

Response: Very good advice. Combined with the comment of reviewer#3, the authors have made detailed changes to the treatment design in the experiment.

  1. Results although presented in detail but require some revision to clarify some of the statements.

Response: Very detailed review. The authors have made a thorough revision of the result analysis, according to the comments of 4 reviewers.

  1. The results have not been discussed properly. It is advisable to discuss the results with the findings of other researchers. Some of the statements are redundant.

Response: Good comment. The authors have made a thorough revision to the discussion and added relevant references to support the explanation of relevant reasons.

  1. The conclusion should be revised and give some recommendations on the basis of their findings.

Response: Good comment. The conclusion has been revised. Such as: No-tillage with wheat straw mulching (NTSM) effectively regulated the dynamics of photosynthetic source of maize with plastic film mulching across the growth period, guaranteed a large leaf area index and duration at the later growth period of maize. Compared to CT control, NTSM increased SPAD, net photosynthetic rate, transpiration rate, and leaf water use efficiency of green leaves in maize plants from VT to R4 stage, which can keep a higher physiological activity at the late growth period of maize. Thereby, NTSM boosted grain yield of maize by 15.6% in comparison to CT, and the high yield was attributed to the synchronous increase and cooperative development of ear number, grain number per ear, and 100-grain weight. This study concluded that no-tillage with straw mulching can be recommended as a promising previous straw management technique for maize production with plastic film mulching to coordinate the conflict between high yield and water scarcity in arid conditions.

  1. The standard of English is up to the mark but it requires improvement.

Response: OK. The authors have made a thorough revision of the text, according to the comments of 4 reviewers.

  1. Some of the references do not conform to the style of the journal.

Response: Very detailed review. According to the basic requirements of the journal, the authors have made a detailed modification of the reference format.

  1. Double-check all the citations in the text to ensure that they are listed in the references list and vice versa.

Response: Very good advice. The authors have checked all references for citations, and added relevant references to support the explanation of relevant reasons.

Round 2

Reviewer 1 Report

This manuscript is a good start to better understanding the crop physiological and yield impacts of no-till and tillage in wheat preceding corn mulched with plastic. However, the co-authors need to make improvements to the manuscript in order for this to be suitable for publication in MDPI Plants. I would be willing to continue to review this manuscript for continuous improvement after the authors have made the following FOUR substantive edits as well as minor line-number specific edits:

    1)      Please accept all track changes and save as another version number. I do not need to see the edits that were made in this draft via track changes or highlighting. Please then make sure there are line numbers which is the default format of the MDPI journal template in Word that you used. I would like to review the manuscript quickly for edits. Thanks!

    2)      Major words in the titles for sub-sections such as the Discussion section need to be all capitalized. 

    3)      At the start of section 2.1., how is it possible the temperature could be 2985°C? 

    4)      The volume(issue) number in the references needs to be in italics. While some journals only use volume numbers, many also have issue numbers and these need to be included. For example, the first reference should be written as:

D'Odorico, P.; Chiarelli, D.D.; Rosa, L.; Bini, A.; Rulli, M.C. The global value of water in agriculture. Proc. Natl. Acad. Sci. U.S.A. 2020, 117, 21985–21993. https://doi.org/10.1073/pnas.2005835117.

Specific Line Number of Manuscript Figure/Table comments (note that requested change of word(s) in quotations where NO edits need to be made for writing before/after each “…”):

Unlike the first draft, for some reason you turned off the line number feature which is automatically set within the MDPI journal Word templates. Please correct this.

Author Response

Comments and Suggestions for Authors

This manuscript is a good start to better understanding the crop physiological and yield impacts of no-till and tillage in wheat preceding corn mulched with plastic. However, the co-authors need to make improvements to the manuscript in order for this to be suitable for publication in MDPI Plants. I would be willing to continue to review this manuscript for continuous improvement after the authors have made the following FOUR substantive edits as well as minor line-number specific edits:

 1) Please accept all track changes and save as another version number. I do not need to see the edits that were made in this draft via track changes or highlighting. Please then make sure there are line numbers which is the default format of the MDPI journal template in Word that you used. I would like to review the manuscript quickly for edits. Thanks!

Response: OK! These problems have been addressed.

 2) Major words in the titles for sub-sections such as the Discussion section need to be all capitalized. 

Response: Very good advice. The format has been modified as required by the MDPI Journal on Plants.

 3) At the start of section 2.1., how is it possible the temperature could be 2985°C? 

Response: OK. This sentence has been deleted, because it did not affect this study.

 4) The volume(issue) number in the references needs to be in italics. While some journals only use volume numbers, many also have issue numbers and these need to be included. For example, the first reference should be written as:

D'Odorico, P.; Chiarelli, D.D.; Rosa, L.; Bini, A.; Rulli, M.C. The global value of water in agriculture. Proc. Natl. Acad. Sci. U.S.A2020117, 21985–21993. https://doi.org/10.1073/pnas.2005835117.

Response: Very good comments. The form of the references has been revised in detail.

Specific Line Number of Manuscript Figure/Table comments (note that requested change of word(s) in quotations where NO edits need to be made for writing before/after each “…”):

Unlike the first draft, for some reason you turned off the line number feature which is automatically set within the MDPI journal Word templates. Please correct this.

Response: Very good comments. The format has been modified as required by the journal.

Round 3

Reviewer 1 Report

This manuscript is a good start to better understanding the crop physiological and yield impacts of no-till and tillage in wheat preceding corn mulched with plastic. The co-authors have made improvements to the manuscript in order for this to be suitable for publication in MDPI Plants. I do not need to review these final edits to the manuscript after the authors have made the following ONE substantive edit as well as minor line-number specific edits:

1)      Remove the period after the word Figure in the manuscript (L185, L206, L222, L242,  L258, L273, and L301).

Specific Line Number of Manuscript Figure/Table comments (note that requested change of word(s) in quotations where NO edits need to be made for writing before/after each “…”):

L45 – Delete “(SPAD)” since this is not used frequently…anywhere else in the manuscript just write SPAD out as “relative chlorophyll content”

L80 – Change (i) to (1)

L81 – Change to “…practices and (2)

L153 – Add (1) to the far right for equation 1

L154 – Add (2) to the far right for equation 2

L155 – Do not indent this line and do not capitalize first word since this is part of the paragraph above

L395 – Start new paragraph with sentence starting with “The high yield…”

L417 – Change to “…NTSM increased relative chlorophyll content, net photosynthetic…”

L436-437 – Between these two section, add Acknowledgements: ???

L440 – Change to “P. Natl. Acad. Sci. U.S.A.

Author Response

Comments and Suggestions for Authors

This manuscript is a good start to better understanding the crop physiological and yield impacts of no-till and tillage in wheat preceding corn mulched with plastic. The co-authors have made improvements to the manuscript in order for this to be suitable for publication in MDPI Plants. I do not need to review these final edits to the manuscript after the authors have made the following ONE substantive edit as well as minor line-number specific edits:

 Remove the period after the word Figure in the manuscript (L185, L206, L222, L242, L258, L273, and L301)

Response: OK! These problems have been addressed.

  L45 – Delete “(SPAD)” since this is not used frequently…anywhere else in the manuscript just write SPAD out as “relative chlorophyll content”. 

Response: Very good advice. It has been revised.

 L80 – Change (i) to (1)

Response: OK. It has been revised.

L81 – Change to “…practices and (2)

Response: OK. It has been revised.

L153 – Add (1) to the far right for equation 1

Response: Very good advice. It has been added.

L154 – Add (2) to the far right for equation 2

Response: Very good advice. It has been added.

L155 – Do not indent this line and do not capitalize first word since this is part of the paragraph above

Response: Very good advice. It has been revised.

L395 – Start new paragraph with sentence starting with “The high yield…”

Response: Very good advice. It has been revised.

L417 – Change to “…NTSM increased relative chlorophyll content, net photosynthetic…”

Response: Very good advice. It has been revised.

L436-437 – Between these two section, add Acknowledgements: ???

Response: Very good advice. It has been added. Such as: Acknowledgments: We would like to thank all the teachers and students of Oasis farming team of Gansu Agricultural University for their help on this study.

L440 – Change to “P. Natl. Acad. Sci. U.S.A.”

Response: Very good advice. It has been revised.

Round 4

Reviewer 1 Report

Thanks! Equations numbers to the right justified all the way

This is NOT written as (Equation 1)

Rather written as (1)